# ANNEALED FISHER IMPLICIT SAMPLER

## ABSTRACT

Sampling from an un-normalized target distribution is an important problem in many scientific fields. An implicit sampler uses a parametric transform $x = G_\theta(z)$ to push forward an easy-to-sample latent code $z$ to obtain a sample $x$. Such samplers are favored for fast inference speed and flexible architecture. Thus it is appealing to train an implicit sampler for sampling from the un-normalized target. In this paper, we propose a novel approach to training an implicit sampler by minimizing the Fisher Divergence between sampler and target distribution. We find that the trained sampler works well for relatively simple targets but may fail for more complicated multi-modal targets. To improve the training for multi-modal targets, we propose another adaptive training approach that trains the sampler to gradually learn a sequence of annealed distributions. We construct the annealed distribution path to bridge a simple distribution and the complicated target. With the annealed approach, the sampler is capable of handling challenging multi-modal targets. In addition, we also introduce a few MCMC correction steps after the sampler to better spread the samples. We call our proposed sampler *the Annealed Fisher Implicit Sampler* (AFIS). We test AFIS on several sampling benchmarks. The experiments show that our AFIS outperforms baseline methods in many aspects. We also show in theory that the added MC correction steps get faster mixing by using the learned sampler as MCMC's initialization.

## 1 INTRODUCTION

Sampling from an un-normalized distribution is an important problem in many scientific fields such as Bayesian statistics (Green, 1995), biology (Schütte et al., 1999), physics simulations (Olsson, 1995), machine learning (Andrieu et al., 2003), and so on. Typically, the problem is formulated as: given a known differentiable un-normalized target potential function $\log p(x)$, one wants to sample from the target distribution. Due to the success of deep neural networks, there is increasing popularity to train a deep generative model to learn to sample(Hu et al., 2018; Wu et al., 2020; Matthews et al., 2022; Corenflos et al., 2021). Such learned models which can approximately sample from target distribution are called samplers.

Training a neural network (i.e., a parameterized transform) $x = G_\theta(z)$ to push forward an easy-to-sample latent code $z \sim p_Z(z)$ to obtain a sample is an appealing approach. Such approaches are favored for fast sampling because they only need a single-time forward pass of neural network transform. Let $G_\theta(.)$ denote the parametric transform and $q(x)$ the un-normalized target distribution with unknown normalizing constant $Z = \int q(x)dx$. Let $p_\theta(x)$ denote the sampler-induced distribution. Some previous work takes a normalizing flow model as sampler, and then minimizes the KL divergence between sampler-induced and target distributions regardless of normalizing constant: $\mathcal{D}_{KL}(p_\theta, q) = \mathbb{E}_{x \sim p_\theta}\big[\log p_\theta(x) - \log q(x) + \log Z\big]$. Note that $Z$ is parameter-free and can be ignored during training. However, minimizing KL divergence relies on explicit log-likelihood of sampler-induced distribution, which can not be computed in a general transform. Such transform with no explicit likelihood is referred to as an implicit sampler.

In this paper, we will focus on implicit samplers. Note that the annoying normalizing constant vanishes when considering the score function of a distribution, $s(x) = \nabla_x \log p(x)$. Thus, we can take the score-based divergence to constructively get rid of the unknown normalizing constant for implicit samplers. Fisher divergence (FD), which is a popular score-based probability divergence, and its variants have obtained much success in recent years, especially in training deep generative models such as energy-based models (Kingma & Cun, 2010; Martens et al., 2012; Song et al., 2019),

score based diffusion models (Song et al., 2020; Kingma et al., 2021; Vahdat et al., 2021; Song & Ermon, 2019; Ho et al., 2020), etc. Assume $p(x), q(x)$ are two probability densities. The Fisher Divergence between $p$ and $q$ is defined as

$$\mathcal{D}_{FD}(p, q) = \frac{1}{2}\mathbb{E}_{x \sim p(x)}\|\nabla_x \log p(x) - \nabla_x \log q(x)\|_2^2.$$

It is always no less than 0 and equals to 0 if and only if $p(x) = q(x)$ a.s. under probability measure $p$. Fisher Divergence is suitable for measuring the dissimilarity between sampler and un-normalized target distribution. So as to be used for training the implicit sampler.

In this paper, we firstly propose a novel approach to learning a sampler by minimizing the Fisher Divergence between sampler and un-normalized target distributions. We call such a sampler the Fisher Implicit Sampler. We then show that the proposed sampler is capable of handling relatively simple target distribution, but would fail for more challenging multi-modal targets.

To remedy this issue and unlock the full potential of the Fisher Implicit Sampler, we additionally propose a novel adaptive training approach that trains the implicit sampler gradually using a sequence of annealed distributions instead of the target distribution. We anneal the target distribution to bridge the hard-to-sample target and an easy-to-sample prior. More precisely, we extend the target distribution $q(x)$ to a sequence of annealed distributions $\{q_k(x)\}_k$ for $k = 0, \ldots, K$, where $q_K(x)$ is the target density and $q_0(x)$ is an easy-to-sample prior distribution, typically a normal distribution. The design of such an annealed path gradually reduces the learning difficulty for the sampler.

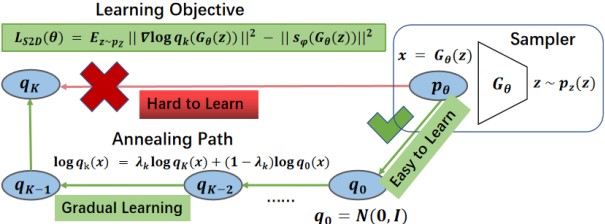

Figure 1: Illustration of proposed **Annealed Fisher Implicit Sampler**.

Moreover, we find that a few steps of MC correction after the sampler help the samples spread better with little cost, as also used in some previous work (Wu et al., 2020; Arbel et al., 2021; Matthews et al., 2022). Combining all together, we call our proposed sampler the Annealed Fisher Implicit Sampler (AFIS), as illustrated in Figure 1. We validate our AFIS on sampling benchmarks, showing improvements over baseline approaches.

The main contributions of our work are summarized as follows:

- We propose a novel loss function to minimize the Fisher Divergence. We show that minimizing the proposed loss is equivalent to minimizing the Fisher Divergence between sampler and target distribution. Note that our objective is largely different from other ones in previous work.

- We provide an insightful understanding of the difficulty in learning multi-modal targets by minimizing Fisher Divergence. We facilitate the annealing technique on training samplers based on our understanding.

- We bring in a novel annealing technique and MC correction steps with our sampler, leading to improved sampling performance with little additional cost.

## 2 BACKGROUND

### 2.1 TRAIN IMPLICIT SAMPLERS WITH SCORE-BASED DIVERGENCE

The learning-to-sample problem arises in many application fields of machine learning. Assume we only have access to an un-normalized target distribution $q(x)$ (or its logarithm $\log q(x)$), and the goal is to approximately sample from the target. In recent years, training a neural network-based transform to approximately sample from target distribution is an appealing method. Such a

transform is called a neural sampler. Let $G_\theta$ denote a neural network which transforms a relatively simple latent code $z \sim p_0(z)$ to a sample $x = G_\theta(z)$. Here, $p_Z(z)$ is an easy-to-sample latent distribution, usually the standard Normal distribution. A general neural sampler does not have an explicit expression of the log-likelihood function, which we name them implicit samplers. Because of the un-normalized target distribution and unavailable log-likelihood, training implicit samplers by minimizing KL or related divergence always fails. An alternative way is to consider score-based divergence.

The Stein Neural Sampler of Hu et al. (2018) is trained by minimizing Stein's Discrepancy between sampler and target distributions. The Stein Discrepancy (SD) (Gorham & Mackey, 2015) is defined as

$$\mathcal{D}_{SD}(p, q) = \sup_{\mathbf{f} \in \mathcal{F}} \left\{ \mathbb{E}_{x \sim p} \langle \nabla_x \log q(x), \mathbf{f}(x) \rangle + \langle \nabla_x, \mathbf{f}(x) \rangle \right\},$$

The calculation of Stein's discrepancy relies on solving a maximization problem w.r.t. test function $\mathbf{f}$. When the function class $\mathcal{F}$ is carefully chosen, the optimal $\mathbf{f}$ may have an explicit solution or easier formulation. For instance, Hu et al. (2018) found that if $\mathcal{F}$ is taken to be $\mathcal{F} = \{\mathbf{f} : \mathbb{E}_p \|\mathbf{f}\|_2^2 \leq \delta\}$, the SD is equivalent to a regularized representation

$$\mathcal{D}_{SD}(p, q) = \max_f \left\{ \mathbb{E}_{x \sim p} \langle \nabla_x \log q(x), \mathbf{f}(x) \rangle + \langle \nabla_x, \mathbf{f}(x) \rangle - \lambda [\mathbf{f}^T \mathbf{f}] \right\}.$$

They used two neural networks: $G_\theta$ to parametrize an implicit sampler and $\mathbf{f}_\eta$ to parametrize the test function. Let $p_\theta(x)$ denote the implicit sampler distribution induced by $x = G_\theta(z)$ with $z \sim p_Z(z)$. Stein Neural Sampler solves a minimax problem on parameter pair $(\theta, \eta)$ to obtain a sampler that minimizes the SD between sampler and target by

$$\min_\theta \max_\eta L(\theta, \eta) = \min_\theta \max_\eta \left\{ \mathbb{E}_{x \sim p_\theta} \langle \nabla_x \log q(x), \mathbf{f}_\eta(x) \rangle + \langle \nabla_x, \mathbf{f}_\eta(x) \rangle - \lambda [\mathbf{f}_\eta^T \mathbf{f}_\eta] \right\}.$$

Here the notion $x \sim p_\theta$ means $x = G_\theta(z)$ with $z \sim p_Z(z)$. They called the above SD the Fisher Stein Discrepancy and the corresponding sampler FSD Neural Sampler.

The Stein Neural Sampler opens the door to training implicit samplers by minimizing score-based Divergence. In fact, the FSD Neural Sampler calculates a surrogate of Fisher Divergence. The FSD's test function $\mathbf{f}$ provides an approximation of Fisher Divergence. However, as we show in Section 3.1, their calculation of Fisher Divergence only provides partial gradient updates of the sampler's parameters, thus leading to training failure even for simple target.

## 2.2 SCORE FUNCTION ESTIMATION

Since the implicit sampler does not have an explicit log-likelihood function or score function, training it with score-based divergence requires inevitably estimating the score function (or equivalent component). Score matching (Hyvärinen & Dayan, 2005) and its variants provided powerful approaches to estimating score function through samples. Assume one only has available samples $x \sim p$, and wants to use a parametric approximated distribution $q_\phi(x)$ to approximate $p$. Such an approximation can be made by minimizing the Fisher Divergence between $p$ and $q_\phi$. We can rewrite the Fisher Divergence as

$$\mathcal{D}_{FD}(p, q_\phi) = \mathbb{E}_{x \sim p} \left\{ \|\nabla_x \log p(x)\|_2^2 + \|\nabla_x \log q_\phi(x)\|_2^2 - 2 \langle \nabla_x \log p(x), \nabla_x \log q_\phi(x) \rangle \right\}.$$

Under certain conditions, the equality $\mathbb{E}_{x \sim p} \langle \nabla_x \log p(x), \nabla_x \log q_\phi(x) \rangle = -\mathbb{E}_{x \sim p} \Delta \log q_\phi(x)$ holds (usually referred to as Stein's Identity(Stein, 1981; Gorham & Mackey, 2017)) . Here $\Delta \log q_\phi(x) = \sum_i \frac{\partial^2}{\partial x_i^2} \log q_\phi(x)$ denotes the Laplacian operator applied on $\log q_\phi(x)$. Combining this equality and noting that the first term of FD $\mathbb{E}_{x \sim p} \|\nabla_x \log p(x)\|_2^2$ does not rely on parameter $\phi$, we have that minimizing $\mathcal{D}_{FD}(p, q_\phi)$ is equivalent to minimizing the following objective

$$\mathcal{L}(\phi) = \mathbb{E}_{x \sim p} \left\{ \|\nabla_x \log q_\phi(x)\|_2^2 + 2 \Delta \log q_\phi(x) \right\}.$$

This objective can be estimated only through samples from $p$, thus is tractable when $q_\phi$ is well-defined. More specifically, one only needs to define a score network $s_\phi(x) \colon \mathbb{R}^D \to \mathbb{R}^D$ instead of a

density to estimate the score function of $p$ in some cases. This technique was proposed in Hyvärinen & Dayan (2005) named after Score Matching. Other variants of score matching were also studied (Song et al., 2019; Vincent, 2011; Pang et al., 2020; Meng et al., 2020; Lu et al., 2022; Bao et al., 2020). Score Matching related techniques have been widely used in training energy-based models and score-based diffusion models in recent years. In this paper, we use score matching related techniques to estimate the score function of the sampler's distribution.

## 3 Annealed Fisher Implicit Sampler

### 3.1 Minimizing the Fisher Divergence: S2D Loss

Let $G_\theta(.)\colon \mathbb{R}^{D_Z} \to \mathbb{R}^{D_X}$ be an implicit sampler (i.e., a neural transform), $p_Z$ latent distribution, $p_\theta$ sampler induced distribution $x = G_\theta(z)$, and $q(x)$ un-normalized target. Our goal is to pull close the FD between $p_\theta$ and $q$ in order to train the sampler. Recall the definition of Fisher Divergence between $p_\theta, q$ is

$$\mathcal{D}_{FD}(p_\theta, q) = \mathbb{E}_{x \sim p_\theta} \|\nabla_x \log p_\theta(x) - \nabla_x \log q(x)\|_2^2.$$

For our learning-to-sample setting, the target score function $\nabla_x \log q(x)$ is known. A direct solution seems work if one uses an additional score network $s_\phi(.)\colon \mathbb{R}^{D_X} \to \mathbb{R}^{D_X}$ to approximate sampler's score function. Samples from implicit sampler is cheap to obtain, so estimating sampler's score function is not hard with score matching related techniques. We call this step the *Score Estimation Step*. With a good approximated $s_\phi(x)$ of sampler's score function, one may wish to minimize the *approximated* Fisher Divergence to update the sampler

$$\theta^* = \arg\min_\theta \mathbb{E}_{x=G_\theta(z), z \sim p_Z(z)} \|s_\phi(x) - \nabla_x \log q(x)\|_2^2.$$

We call this step the *Score Difference Minimization Step*. By alternating the above two steps, one may wish the Fisher divergence will be minimized, thus the training of sampler is done. We name the resulting approach the *Direct Method*. Interestingly, the *Direct Method* coincides with FSD Neural Sampler as we state in Proposition 1. We put detailed proof in Appendix A due to limited pages.

**Proposition 1.** *Estimating the sampler's score function $s_\phi(.)$ with score matching is equivalent to maximizing the Fisher Stein Discrepancy objective to obtain FSD's optimal test function. More specially, the optimal score estimation $s^*$ and FSD optimal test function $\mathbf{f}^*$ satisfy*

$$\mathbf{f}^*(x) = \frac{1}{2\lambda}\big[\nabla_x \log q(x) - s^*(x)\big].$$

*Moreover, the Direct method is equivalent to FSD when training implicit Sampler.*

**Although the direct method seems reasonable, it fails as we show in the experiment on a simple Banana target in Figure 2.** We find that even if sampler's score function is estimated perfectly at each iteration, the direct method still gives only partial parameter gradient for minimizing the Fisher Divergence. We start by analyzing Fisher Divergence's gradient w.r.t. sampler's parameter. The Fisher Divergence is

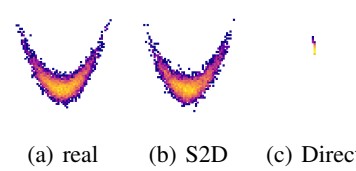

(a) real     (b) S2D     (c) Direct

Figure 2: Direct method fails for simple Banana distribution while S2D loss succeeds.

$$\mathcal{L}_{FD}(\theta) = \mathbb{E}_{x \sim p_\theta} \|\nabla_x \log q(x) - \nabla_x \log p_\theta(x)\|_2^2.$$

One wants to adjust $\theta$ to minimize $\mathcal{L}_{FD}(\theta)$. The $\theta$ gradient of the above objective writes

$$\frac{\partial}{\partial \theta} \mathbb{E}_{p_\theta} \|s_d(x) - s_\theta(x)\|^2 = \mathbb{E}_{p_\theta} \|s_d(x) - s_\theta(x)\|^2 \frac{\partial}{\partial \theta} \log p_\theta(x) + \mathbb{E}_{p_\theta} 2(s_\theta(x) - s_d(x))^T \frac{\partial}{\partial \theta} s_\theta(x).$$

The first gradient term coincides with the direct approach if we asynchronously estimate the sampler's score function perfectly. More precisely, with perfect score estimation $s_\phi(x) = \nabla_x \log p_\theta(x)$, we have

$$\frac{\partial}{\partial \theta} \mathbb{E}_{x \sim p_\theta} \|\nabla_x \log q(x) - s_\phi(x)\|_2^2$$

$$= \frac{\partial}{\partial \theta} \int \|\nabla_x \log q(x) - s_\phi(x)\|_2^2 p_\theta(x) dx = \int \|\nabla_x \log q(x) - s_\phi(x)\|_2^2 \frac{\partial}{\partial \theta} p_\theta(x) dx$$

$$= \int \|\nabla_x \log q(x) - s_\phi(x)\|_2^2 p_\theta(x) \frac{\partial}{\partial \theta} \log p_\theta(x) dx = \mathbb{E}_{x \sim p_\theta} \|\nabla_x \log q(x) - s_\phi(x)\|_2^2 \frac{\partial}{\partial \theta} \log p_\theta(x).$$

The above equation reveals that the direct method only takes partial gradient to minimize the FD between sampler and target. In many cases, this partial gradient leads to training failure as we observe in Figure 2. In (Hu et al., 2018), FSD Neural Sampler used Kernelized Stein Discrepancy trained implicit sampler as initialization before training with FSD. However, such initialization limits the usage of FSD because the optimization might start from a local minima which is close to KSD's local minima and can potentially be mislead the sampler.

In order to minimize the Fisher Divergence correctly, we propose a novel training objective called *Score Square Difference* loss (S2D) which accounts for the full parameter gradient to minimize the Fisher Divergence. The S2D loss is defined as the difference of target and sampler's square score norm, where the sampler's score function is estimated asynchronously with a score network $s_\phi(x)$. More precisely, our S2D loss is defined as

$$\mathcal{L}_{S2D}(\theta) := \mathbb{E}_{x \sim p_\theta}\left\{\|\nabla_x \log q(x)\|_2^2 - \|s_\phi(x)\|_2^2\right\},$$

where $s_\phi(.)$ is the estimated score function of sampler distribution. The score function is usually estimated by score matching related techniques. The notation $x \sim p_\theta$ means $x = G_\theta(z), z \sim p_Z(z)$. The following proposition 2 shows that, if the sampler score function is estimated perfectly, the parameter gradient of S2D loss is the same as the gradient of Fisher Divergence.

**Proposition 2.** *Assume $s_\phi(x) = \nabla_x \log p_\theta(x)$. Then the following equality holds:*

$$\frac{\partial}{\partial \theta}\mathcal{L}_{S2D}(\theta) = \frac{\partial}{\partial \theta}\mathcal{L}_{FD}(\theta).$$

We give the detailed proof in Appendix B. **This proposition says that, if we alternate between score estimation of sampler's score function, and minimization of the S2D loss, we are actually minimizing the Fisher Divergence between sampler and target**. The S2D loss is a surrogate of Fisher Divergence which can provide the same parameter gradient as Fisher Divergence. So minimizing the S2D loss gives the same results as minimizing the intractable Fisher Divergence. Figure 3 gives an illustration of the relation between S2D loss and Fisher Divergence. The black curve stands for the intractable Fisher Divergence. Green curve represents the S2D loss. The S2D loss shares the same gradient parameter as Fisher Divergence. We refer to a sampler trained with such approach the *Fisher Implicit Sampler* (FIS). We give an algorithm for FIS in **Algorithm** 1. We take standard score matching as an illustration of score estimation step, but other score estimation techniques such as denoising score matching and sliced score matching also works.

---

**Algorithm 1:** Fisher Implicit Sampler training

---

**Input:** un-normalized target $\log q(x)$, latent distribution $p_Z(z)$, implicit sampler $G_\theta$, score network $s_\phi$, mini-batch size B, max iteration M.

Randomly initialize $(\theta^{(0)}, \phi^{(0)})$.

**for** $t$ *in 0:M* **do**

    # *update score network parameter*

    Get mini-batch $x_i = G_{\theta^{(t)}}(z_i), z_i \sim p_Z(z), i = 1, .., B$.

    Calculate score matching objective:

    $\mathcal{L}_{SM}(\phi) = \frac{1}{B}\sum_{i=1}^{B}\left[\|s_\phi(x_i)\|_2^2 + 2\langle\nabla_x, s_\phi(x_i)\rangle\right]$.

    Minimize $\mathcal{L}_{SM}(\phi)$ to get $\phi^{(t+1)}$.

    # *update sampler parameter*

    Get mini-batch latent code $z_i \sim p_Z(z), i = 1, \ldots, B$.

    Use re-parametrization trick to calculate S2D loss for sampler

    $\mathcal{L}_{S2D}(\theta) = \frac{1}{B}\sum_{i=1}^{B}\left[\|\nabla_x \log q(G_\theta(z_i))\|_2^2 - \|s_{\phi^{(t+1)}}(G_\theta(z_i))\|_2^2\right]$.

    Minimize $\mathcal{L}_{S2D}(\theta)$ to get $\theta^{(t+1)}$.

**end**

**return** $(\theta, \phi)$.

---

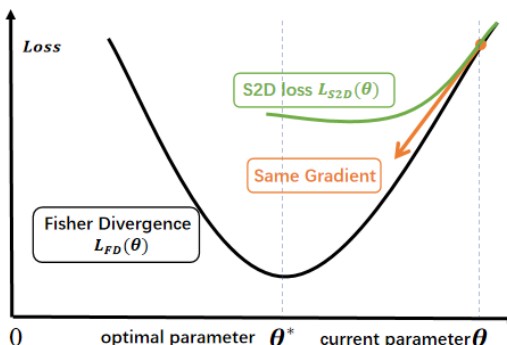

Figure 3: S2D loss and Fisher Divergence. The S2D loss shares the same parameter gradient as Fisher Divergence if sampler's score is estimated perfectly asynchronously. Thus minimizing the S2D loss to update the sampler is equivalent to minimizing the Fisher Divergence between sampler and target.

Figure 2 shows that our proposed FIT (S2D loss) can successfully train an implicit sampler from scratch to sample from the famous banana shape distribution. While the Direct method fails to train the correct sampler.

Although FIT is capable of handling benchmark targets, we find that FIT fails on more challenging multi-modal targets with very separated modes. To remedy the multi-modal failure issues and fully unlock the potential of the S2D loss, we propose to combine the annealing techniques with FIT for multi-modal targets. The idea of annealing is widely used in sampling and stochastic optimization literature (Neal, 2001; Salimans et al., 2015; Chen et al., 2016; Doucet et al., 2001; Van Laarhoven & Aarts, 1987). The technique constructs a distribution bridge between a relatively simple prior distribution and a complicated target. The learning (or other operations such as sampling or optimization) are gradually operated on each middle distribution from prior to the target. Typically, the annealing technique can lower the barrier of operation of the target by dispersing the difficulty to all middle distributions.

## 3.2 ANNEALED FISHER IMPLICIT TRAINING

By executing FIT steps repeatedly, the sampler is trained to minimize the Fisher divergence between $p_\theta$ and target $q$. However, directly minimizing the Fisher divergence is problematic in practice. If the sampler's distribution is too dissimilar to the target, the Fisher divergence could be hard to estimate accurately as mentioned in (Wenliang & Kanagawa, 2020). The Fisher divergence can be small under any tolerance even if two distribution are largely different in terms of KL divergence. More precisely, the Fisher Divergence is likely inaccurate if two distributions are too dissimilar. Due to this issue, the sampler might not be able to estimated the Fisher divergence accurately, making the training fail. In fact, above issue occurs a lot in real applications. Sampler is often initialized to concentrate around the origin, while the target distribution rarely concentrates around the origin.

To remedy the inaccurate Score Estimation issue, we need to guide the sampler to start from learning a relatively simple target, and then the more challenging one. Based on such intuition, we introduce a gradual relaxation of target distribution. More precisely, we construct a sequence of annealed distributions $\{q_k\}, k \in \{0,..,K\}$ which gradually transform a relatively simple distribution $q_0$ to target distribution $q_K = q$. Typically, $q_0$ is chosen as $\mathcal{N}(\mathbf{0}, \mathbf{I})$ for simplicity. We let the sampler gradually learn to sample from each $q_k$ with $k$ increasing from $k = 0$ to $k = K$. Since when $k$ is small $q_k$ is simpler than $q_K$, the estimation of Fisher divergence is easier. Thus the sampler can learn to approximate $q_k$. When one gradually turns $k$ to $k = K$, the sampler will gradually learn to sample from our final target $q_K = q$.

Such easy-to-hard technique is commonly known as **annealing techniques**. Marinari & Parisi (1992); Geyer & Thompson (1995) proved faster mixing time with temperature annealed target. Wenzel et al. (2020) utilized anneal path to connect model and posterior in Bayesian inference regime. Mandt et al. (2016); Huang et al. (2018); Fu et al. (2019) annealed the KL regularization in variational inference. D'Angelo & Fortuin (2021) proposed to anneal the target when running Stein Variational Gradient Descent algorithm for better mixing speed. Perhaps the most similar annealing approach to ours is Neal (2001); Wu et al. (2020) which construct a geometric distribution path $p_k(x)$ between a Gaussian prior and target density. We utilize a similar anneal path as in Wu et al. (2020).

In this paper, we anneal the target distribution $q$ with a geometric interpolation starting with a standard Gaussian distribution as prior

$$\log q_k(x) = \lambda_k \log q_K(x) + (1 - \lambda_k) \log q_0(x)$$

with $q_0 = \mathcal{N}(\mathbf{0}, \mathbf{I})$ and $0 \leq \lambda_k \leq 1$ a pre-defined annealing schedule function with $\lambda_0 = 0, \lambda_K = 1$. The score function is then linearly interpolated with

$$\nabla_x \log q_k(x) = \lambda_k \nabla_x \log q_K(x) + (1 - \lambda_k) \nabla_x \log q_0(x),$$

where $\nabla_x \log q_K(x) = \nabla_x \log q(x)$ is the target score function and $\nabla_x \log q_0(x)$ the prior score. For standard Normal prior, we have $\nabla_x \log q_0(x) = -x$. We name our FIS sampler combined with annealing technique the *Annealed Fisher Implicit Training*. Because of the pages limitation, we put the full AFIS algorithm in Appendix F. By annealing the target distribution to a sequence of easier-to-learn targets, we divide the difficulty of sampler to learn one final distribution to learn sequentially from less difficult targets. Thus the sampler will not be bothered by inaccurate Fisher divergence estimation and training failure. Figure 1 gives a brief summary of how AFIS works. The Annealed Fisher Implicit Sampler is trained along annealed distributions progressively.

### 3.3 MONTE CARLO CORRECTION

Deterministic sampler suffers from mode-connection issue. The issue says that a deterministic transform can not fully disconnect two modes as studied in Wu et al. (2020). Such issue limit the use of a pure deterministic sampler. Recent works show that combining stochastic corrections with deterministic transforms could improve the sampling performance (Wu et al., 2020; Song et al., 2020; Song & Ermon, 2019). MCMC(Hastings, 1970; Roberts & Rosenthal, 1998; Xifara et al., 2014; Neal, 2011) is a commonly used stochastic transform family. By running MCMC, one can approximated sample from some un-normalized target distribution. Thus a few steps MCMC is a nice way to serve as stochastic corrections.

In particular, after training the sampler, we take the generated samples $x = G_\theta(z), z \sim p_0(z)$ as initialization and run several MCMC as correction steps to spread samples for better diversity. Both energy-based and score-based MCMC can be used. We take the Langevin MC as an illustration and put more details of MC corrections in Appendix C. Note that our method is not limited to these MC corrections.

**Langevin Dynamic Correction** A set of particles is assumed to reach $q(x)$ as a stationary distribution if it is driven by a Langevin Dynamic with local updates

$$dX_t = \nabla_{X_t} \log q(X_t)/2 + dW_t,$$

where $W_t$ is standard Brownian motion. The discrete scheme of Langevin Correction is given by

$$X^{(t+1)} = X^{(t)} + \frac{\epsilon}{2} \nabla \log q(X^{(t)}) + \sqrt{\epsilon} Z^{(t)},$$

where $Z^{(t)} \sim \mathcal{N}(0; I)$. The Fokker-Planck equation tells that under certain conditions, $q(x)$ is the only stationary distribution of above diffusion dynamic. About 20 updates of steps is sufficient to have good enough correction effects in practice.

The combination of deterministic sampler and stochastic correction in fact gives faster mixing for MCMC. The deterministic sampler sample particles coarsely near target's high density modes. After that, the MCMC helps the particle spread better around each modes. In particular, we show that Langevin mixing time can be controlled by Fisher divergence between sampler distribution and target. Taking advantage of flexible neural network architecture, AFIS can be trained to match target score at any precision. The Theorem 1 shows that Langevin Correction's mixing time can be reduced by well trained sampler.

**Theorem 1.** *Assume the target potential* $\log q(x)$ *is smooth and satisfies*

$$\lim_{\|x\|_2 \to +\infty} \left( \frac{\|\nabla \log q(x)\|_2^2}{2} - \Delta \log q(x) \right) = +\infty.$$

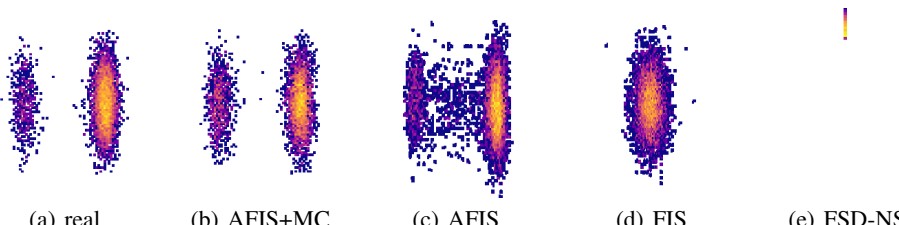

(a) real      (b) AFIS+MC      (c) AFIS      (d) FIS      (e) FSD-NS

Figure 4: Sample comparison on Double Well targets. (a) real samples; (b) samples from trained AFIS with 5 steps of HMC correction; (c) samples from trained AFIS; (d) samples from trained FIS without annealing; (e) samples from trained FSD-NS. All samplers and score networks use the same architecture.

*Assume generated distribution $p$ induced by AFIS $x = G(z)$ is trained to match Fisher divergence under $\delta$ precision $\mathcal{D}_F(p,q) \leq \delta$. Then there exists a positive constant $\lambda$ and a dimension-free positive constant $C$ which only depend on target distribution $q(x)$, such that under Langevin diffusion with initial distribution $p_0 = p$,*

$$dX_t = \nabla \log q(x)/2dt + dW_t,$$

*the diffusion time*

$$T^* = \max\left\{0, \frac{1}{2\lambda}\big[C + \log(\frac{\delta}{\epsilon})\big]\right\}$$

*is enough to control the KL divergence between corrected distribution $p_T$ and target $q$ under tolerance $\epsilon$.*

In practice, the AFIS can be trained to achieve any precision to match target under Fisher Divergence. The above theorem says, the better AFIS is trained, the shorter time for MC correction is needed to achieve same tolerance in terms of KL divergence. We provide the detailed proof in Appendix D.

### 3.4 COMBINING ALL: THE ANNEALED FISHER IMPLICIT SAMPLER

Combining the S2D loss, the annealed technique, and MC corrections, we obtain our final sampler: the Annealed Fisher Implicit Sampler (AFIS) with MC corrections. Figure 4 shows a comparison of trained sampler's samples on Double Well distribution. Double Well is a usually used bi-variate testing target with two separated modes. The figure shows that **the AFIS with a few steps of MC correction gives the best samples**. The AFIS with no MCMC correction can not fully separate two disjoint modes. The FIS (without annealing) fails to learn the two modes. The FSD-NS (or the Direct Method) also fails for training. To be concluded, the experiments show that S2D loss, annealed technique, and MC correction all contribute to successful learning.

## 4 EXPERIMENTS

### 4.1 AFIS FOR SYNTHETIC TARGET

For sanity check, we apply AFIS on some toy target distributions as used in Hu et al. (2018); Rezende & Mohamed (2015). The anneal path $p_\lambda(x) \propto \exp(\lambda \log p_{target}(x) + (1 - \lambda) \log p_{prior}(x))$ starts from a Normal distribution when $\lambda = 0$ and ends with the target when $\lambda = 1$. Let $M$ be the number of max iterations, and $t$ be the current training iteration. We set $\lambda_i$ to grow linearly from 0 to 1 when $i < 9M/10$. We train the sampler with real target $\log q(x)$ for rest $M/10$ iterations. The annealed path reduces the bar of learning to sample, resulting relatively accurate updating direction for the current sampler. The sampler is guided along the annealed path towards the target. We defer the detailed experiment settings and more results to Appendix E.1.

Specifically, we visualize the sample results on three distributions with hard-to-sample characteristics such as multi-modality and periodicity, as shown in Figure 5. It shows that samples from our AFIS+MC method perfectly match all target distributions. For quantitative comparison, we calculate the Maximum Mean Discrepancy between the pure HMC samples and all samplers' samples. The FSD-NS does not converge when training, so we omit the result of FSD-NS in comparison. Since the task focuses on training implicit samplers, we do not compare other explicit samplers. Table 1 summarizes the results of the MMD evaluation of all samplers. In all datasets, our AFIS consistently performs better than FIS. With additional MC correction steps, we always get lower MMD compared to the pure AFIS method.

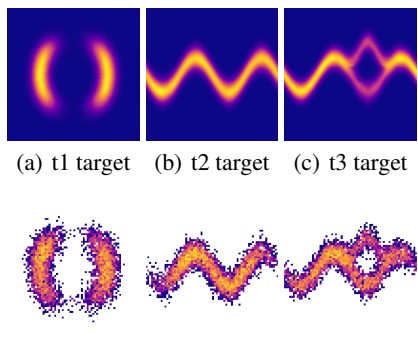

(a) t1 target   (b) t2 target   (c) t3 target

(d) t1 sample  (e) t2 sample  (f) t3 sample

Figure 5: Target and AFIS+MC samples.

Table 1: MMD (with rbf kernel) evaluation for synthetic targets. Additional 10 Langevin MC correction steps are used in AFIS+MC sampler. The lower the metric, the better the sampler.

| Target | banana | double well | t1 | t2 | t3 |
|---|---|---|---|---|---|
| FIS(ours) | 1.12e-2±1.07e-3 | 3.51e-1±3.06e-3 | 4.54e-2±4.10e-3 | 7.48e-2±1.81e-3 | 5.18e-2±2.68e-3 |
| AFIS(ours) | 7.07e-4±1.72e-4 | 1.07e-2±1.37e-3 | 3.31e-3±1.13e-3 | 4.64e-2±2.53e-3 | 2.65e-2±1.91e-3 |
| AFIS+MC(ours) | **2.45e-4±1.20e-4** | **5.99e-3±1.33e-3** | **2.15e-3±8.37e-4** | **3.61e-2±2.67e-3** | **2.20e-2±1.97e-3** |

## 4.2 BAYESIAN REGRESSION

We also test our Implicit Sampler on Bayesian regression tasks as in Song et al. (2017). HMC is a good baseline for such tasks, as pointed out in Neklyudov et al. (2020); Neklyudov & Welling (2022). The inference of the Bayesian logistic regression model aims to sample from the posterior distribution. We compare FIS (no anneal), AFIS, and AFIS+MC on Australian, German, and Heart datasets. To evaluate samples' quality, we run HMC as a baseline to obtain approximated samples from target distributions and calculate Maximum Mean Discrepancy between samples from implicit samplers and HMC baseline. Table 2 shows the results of the Bayesian inference experiments. Other than FSD-NS, which always fails during training, our generators can generate high-quality samples. Moreover, annealed technique and MC correction steps further improve sample quality. Experimental details can be found in Appendix E.2.

Table 2: MMD (with rbf kernel) evaluation for posterior sampling. Additional 10 Langevin MC correction steps are used in AFIS+MC sampler. The lower the metric, the better the sampler.

| Posterior | Australian | German | Heart |
|---|---|---|---|
| FIS(ours) | 7.99e-3±2.81e-4 | 1.91e-4±6.48e-6 | 9.84e-5±1.08e-5 |
| AFIS(ours) | 6.30e-3±2.50e-4 | **2.42e-6±4.02e-7** | 3.66e-5±1.08e-5 |
| AFIS+MC(ours) | **2.16e-3±1.08e-4** | 2.46e-6±3.97e-7 | **3.64e-5±1.07e-5** |

## 5 CONCLUSION

We have presented a novel approach for training an implicit sampler to sample from un-normalized density. Our approach minimizes the Fisher Divergence with the aid of an asynchronous score network. We show theoretically that our method can accurately minimize the Fisher Divergence for the implicit sampler, which is the first one as far as we know. Besides, our approach uses both the annealing technique and stochastic corrections for improved sampling performance. We also prove the faster mixing for MC correction. We test our approach on commonly used synthetic target generation and Bayesian regression benchmarks and observe ideal performance.

ETHICS STATEMENT

Our work proposes an approach to train an implicit sampler by minimizing Fisher Divergence between sampler and target distribution. Since the research is a fundamental methodology in machine learning, the negative consequences of the methodology seem not obvious.

REPRODUCIBILITY STATEMENT

We provide details of our approach and sampler in Appendix. We provide complete proofs of all theoretical results also in Appendix. We also propose the python code for implementation. We state that our research is reproducible.

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

# A PROOF OF PROPOSITION 1

We provide the proof of Proposition 1 here.

*Proof.* With fixed $p$ and known target $q$, the optimal test function $\mathbf{f}^*$ has representation

$$\mathbf{f}^* = \arg\min_{\mathbf{f}} \mathcal{L}(\mathbf{f})$$

Where functional $\mathcal{L}(\mathbf{f})$ has integral representation

$$\mathcal{L}(f) = \mathbb{E}_{x \sim p} \Big\{ \langle \nabla_x \log q(x), \mathbf{f}(x) \rangle + \langle \nabla_x, \mathbf{f}(x) \rangle - \lambda[\mathbf{f}^T(x)\mathbf{f}(x)] \Big\}$$

$$= \int p(x)\langle \nabla_x \log q(x), \mathbf{f}(x) \rangle + p(x)\langle \nabla_x, \mathbf{f}(x) \rangle - \lambda p(x)[\mathbf{f}^T(x)\mathbf{f}(x)]dx$$

$$= \int l(x, \mathbf{f}, \nabla \mathbf{f})dx.$$

Here $l(x, \mathbf{f}, \nabla\mathbf{f}) = \int p(x)\langle \nabla_x \log q(x), \mathbf{f}(x) \rangle + p(x)\langle \nabla_x, \mathbf{f}(x) \rangle - \lambda p(x)[\mathbf{f}^T(x)\mathbf{f}(x)]$. By Euler-Lagrange equation, the optimal function $\mathbf{f}$ satisfies

$$\frac{\partial l}{\partial \mathbf{f}} - \frac{d}{dx}\Big(\frac{\partial l}{\partial \mathbf{f}'}\Big) + \frac{\partial^2}{\partial x^2}\Big(\frac{\partial l}{\partial \mathbf{f}''}\Big) = 0.$$

By calculation, we have

$$\frac{\partial l}{\partial \mathbf{f}}(x) = p(x)\nabla \log q(x) - 2\lambda p(x)\mathbf{f}(x)$$

$$\frac{d}{dx}\Big(\frac{\partial l}{\partial \mathbf{f}'}\Big)(x) = \nabla_x p(x)$$

$$\frac{\partial l}{\partial \mathbf{f}''}(x) = 0.$$

So the optimal $\mathbf{f}^*$ satisfies the Euler-Lagrange equation as

$$p(x)\nabla_x \log q(x) - 2\lambda p(x)\mathbf{f}(x) - \nabla_x p(x) = 0.$$

Divide the both side with $p(x)$ and note that $\nabla_x p(x)/p(x) = \nabla_x \log p(x)$, the equation turns to

$$\mathbf{f}^*(x) = \frac{1}{2\lambda}\big[\nabla_x \log q(x) - \nabla_x \log p(x)\big].$$

Next consider optimal $s^*$. The $s^*$ is obtained by minimizing the Score Matching objective, which is equivalent to minimizing the Fisher divergence between $p$ and $s$ induced family, thus the optimal $s^*(x) = \nabla_x \log p(x)$. Substitute $\nabla_x \log p(x)$ with $s^*$ into $f^*$ formula, we have

$$\mathbf{f}^*(x) = \frac{1}{2\lambda}\big[\nabla_x \log q(x) - s^*(x)\big].$$

$\square$

# B PROOF OF PROPOSITION 2

In this section, we prove that the S2D loss and Fisher Divergence shares exactly the same parameter gradient.

*Proof.* Let $p_\theta$ denote sampler's distribution. $s_\theta$ denote the true but unknown sampler's score function. $q$ denotes the known un-normalized target. For rest of the proof, the notion $\|x\|$ represents the $L^2$ norm of a vector in $D_X$ dimensional Euclidean space $x \in \mathbb{R}^{D_X}$. Recall that the Fisher Divergence is defined as

$$\mathcal{L}_{FD}(\theta) = \mathbb{E}_{x \sim p_\theta}\|\nabla_x \log q(x) - s_\theta(x)\|_2^2.$$

Thus the sampler parameter gradient of Fisher Divergence writes

$$
\frac{\partial}{\partial \theta} \mathbb{E}_{p_\theta} \|\nabla_x \log q(x) - s_\theta(x)\|^2 = \frac{\partial}{\partial \theta} \int \|\nabla_x \log q(x) - s_\theta(x)\|_2^2 p_\theta(x) dx
$$

$$
= \int \|\nabla_x \log q(x) - s_\theta(x)\|_2^2 \frac{\partial}{\partial \theta} p_\theta(x) dx + \int p_\theta(x) \frac{\partial}{\partial \theta} \|\nabla_x \log q(x) - s_\theta(x)\|_2^2 dx
$$

$$
= \mathbb{E}_{p_\theta} \|\nabla_x \log q(x) - s_\theta(x)\|^2 \frac{\partial}{\partial \theta} \log p_\theta(x) + \mathbb{E}_{p_\theta} 2(s_\theta(x) - \nabla_x \log q(x))^T \frac{\partial}{\partial \theta} s_\theta(x)
$$

$$
= (1) + (2).
$$

The first term can be estimated with

$$
(1) = \int \|\nabla_x \log q(x) - s_\theta(x)\|^2 \frac{\partial}{\partial \theta} p_\theta(x) dx
$$

$$
= \frac{\partial}{\partial \theta} \int \mathbf{sg}\big[\|\nabla_x \log q(x) - s_\theta(x)\|^2\big] p_\theta(x)
$$

$$
= \frac{\partial}{\partial \theta} \mathbb{E}_{p_\theta} \mathbf{sg}\Big[\|\nabla_x \log q(x) - s_\theta(x)\|^2\Big].
$$

Here the operator $\mathbf{sg}$ denotes **stop gradient operator** with respect to parameter $\theta$. $\mathbf{sg}[f_\theta]$ stop the parameter dependence of $\theta$ for function $f$, meaning that one can only evaluate $f_\theta(x)$ point-wise but can not obtain the $\theta$ gradient of $f_\theta(x)$. Here we stop the gradient of function $\|\nabla \log q(x) - s_\theta(x)\|^2$, so we can use another score network $s_\phi$ to approximate $s_\theta$ point-wise, regardless of the $\theta$ parameter dependence. Next we consider the second term. The second term turns to

$$
(2) = \mathbb{E}_{p_\theta} 2(s_\theta(x) - \nabla_x \log q(x))^T \frac{\partial}{\partial \theta} s_\theta(x)
$$

$$
= \mathbb{E}_{p_\theta} 2(s_\theta(x) - \nabla_x \log q(x))^T \frac{\partial}{\partial \theta} \nabla_x \log p_\theta(x)
$$

$$
= 2 \int p_\theta(x)(s_\theta(x) - \nabla_x \log q(x))^T \frac{\partial}{\partial \theta} \frac{\partial}{\partial x} \log p_\theta(x) dx
$$

$$
= 2 \int p_\theta(x)(s_\theta(x) - \nabla_x \log q(x))^T \frac{\partial}{\partial \theta} \left[\frac{1}{p_\theta(x)} \frac{\partial p_\theta(x)}{\partial x}\right] dx
$$

$$
= 2 \int (s_\theta(x) - \nabla_x \log q(x))^T \left[\frac{\partial}{\partial \theta} \frac{\partial}{\partial x} p_\theta(x)\right] dx - 2 \int p_\theta(x)(s_\theta(x) - \nabla_x \log q(x))^T \left[\frac{\partial \log p_\theta(x)}{\partial x} \frac{\partial \log p_\theta(x)}{\partial \theta}\right]
$$

$$
= (3) + (4).
$$

Looking at $(3)$, we have

$$
(3) = 2 \int (s_\theta(x) - \nabla_x \log q(x))^T \left[\frac{\partial}{\partial \theta} \frac{\partial}{\partial x} p_\theta(x)\right] dx
$$

$$
= 2 \int \frac{\partial}{\partial \theta} \left\{\mathbf{sg}\Big[(s_\theta(x) - \nabla_x \log q(x))\Big]^T \frac{\partial}{\partial x} p_\theta(x)\right\} dx
$$

$$
= 2 \frac{\partial}{\partial \theta} \int \frac{\partial}{\partial \epsilon} p_\theta(x + \epsilon v) dx, \quad v = \mathbf{sg}\Big[(s_\theta(x) - \nabla_x \log q(x))\Big], \epsilon = 0
$$

$$
= 2 \frac{\partial}{\partial \theta} \frac{\partial}{\partial \epsilon} \int p_\theta(x + \epsilon v) dx
$$

$$
= 2 \frac{\partial}{\partial \theta} \frac{\partial}{\partial \epsilon} \mathbf{1}
$$

$$
= 0.
$$

Above equality holds because of $\int p_\theta(x + \epsilon v) dx = 1$ holds for all $v, \theta, \epsilon$. If we view $\epsilon$ as a shift strength parameter, the above equality recovers the first order Bartlett identity (Bartlett, 1953).

Next we turns to term (4). Note that

$$
\begin{aligned}
(4) &= -2 \int p_\theta(x)(s_\theta(x) - \nabla_x \log q(x))^T \left[ \frac{\partial \log p_\theta(x)}{\partial x} \frac{\partial \log p_\theta(x)}{\partial \theta} \right] \\
&= -2 \int p_\theta(x) \left[ (s_\theta(x) - \nabla_x \log q(x))^T \frac{\partial \log p_\theta(x)}{\partial x} \right] \frac{\partial \log p_\theta(x)}{\partial \theta} \\
&= -2 \int \left[ (s_\theta(x) - \nabla_x \log q(x))^T \frac{\partial \log p_\theta(x)}{\partial x} \right] \frac{\partial p_\theta(x)}{\partial \theta} \\
&= -2 \frac{\partial}{\partial \theta} \int \mathbf{sg} \left[ (s_\theta(x) - \nabla_x \log q(x))^T \frac{\partial \log p_\theta(x)}{\partial x} \right] p_\theta(x) \\
&= -2 \frac{\partial}{\partial \theta} \int \mathbf{sg} \left[ (s_\theta(x) - \nabla_x \log q(x))^T \frac{\partial \log p_\theta(x)}{\partial x} \right] p_\theta(x) \\
&= -2 \frac{\partial}{\partial \theta} \mathbb{E}_{p_\theta(x)} \mathbf{sg} \left[ (s_\theta(x) - \nabla_x \log q(x))^T \frac{\partial \log p_\theta(x)}{\partial x} \right] \\
&= -2 \frac{\partial}{\partial \theta} \mathbb{E}_{p_\theta(x)} \left\{ \mathbf{sg} \left[ (s_\theta(x) - \nabla_x \log q(x)) \right]^T \mathbf{sg} \left[ \frac{\partial \log p_\theta(x)}{\partial x} \right] \right\} \\
&= -2 \frac{\partial}{\partial \theta} \mathbb{E}_{p_\theta(x)} \left\{ \mathbf{sg} \left[ (\nabla_x \log q(x) - s_\theta(x)) \right]^T \mathbf{sg} \left[ s_\theta(x) \right] \right\}.
\end{aligned}
$$

Combining all above, we calculate the parameter derivative as

$$
\begin{aligned}
&\frac{\partial}{\partial \theta} \mathbb{E}_{p_\theta} \| \nabla_x \log q(x) - s_\theta(x) \|^2 \\
&= (1) + (2) = (1) + (3) + (4) \\
&= \frac{\partial}{\partial \theta} \mathbb{E}_{p_\theta} \mathbf{sg} \left[ \| \nabla_x \log q(x) - s_\theta(x) \|^2 \right] + 0 - 2 \frac{\partial}{\partial \theta} \mathbb{E}_{p_\theta(x)} \left\{ \mathbf{sg} \left[ (s_\theta(x) - \nabla_x \log q(x)) \right]^T \mathbf{sg} \left[ s_\theta(x) \right] \right\} \\
&= \frac{\partial}{\partial \theta} \mathbb{E}_{p_\theta} \left\{ \mathbf{sg} \left[ \| \nabla_x \log q(x) \|^2 \right] - \mathbf{sg} \left[ \| s_\theta(x) \|^2 \right] \right\}.
\end{aligned}
$$

Thus the equivalent loss function

$$
\mathcal{L}_{S2D}(\theta) = \mathbb{E}_{p_\theta} \left\{ \mathbf{sg} \left[ \| \nabla_x \log q(x) \|^2 \right] - \mathbf{sg} \left[ \| s_\theta(x) \|^2 \right] \right\}.
$$

Share the same parameter gradients as the Fisher divergence which is intractable. Since we only need the $x$ gradient of sampler score function $s_\theta$ (because the stop gradient operator), so we can estimate $s_\theta(x)$ through another score network $s_\phi(x)$ with samples consistently obtained from sampler. With above objective function, we could minimize the Fisher divergence between $p_\theta$ and $q$. $\qquad\square$

## C  INTRODUCTION TO METROPOLIS-HASTINGS AND HAMILTONIAN CORRECTION

Assume the target distribution is $p(x)$, the MH MCMC requires a proposal distribution $p(\tilde{x}|x)$ to propose candidate samples $\tilde{x} \sim q(\tilde{x}|x)$. The Markov chain then accept the candidate sample with probability $r = \min\{\frac{p(\tilde{x})q(x|\tilde{x})}{p(x)q(\tilde{x}|x)}, 1\}$. Under some conditions, the chain will eventually reach $p(x)$ as stationary distribution. The proposal distribution can be symmetric or non-symmetric. Conditional gaussian $q(\tilde{x}|x) = \mathcal{N}(x; \sigma^2)$ is a usual choice. Proposals based on score function $q(\tilde{x}|x) = \mathcal{N}(x + \frac{\epsilon}{2}\nabla_x \log p(x), \sigma^2)$ is also popular (Xifara et al., 2014). If one consider an auxiliary state space of $(x, v)$ and execute the proposal in such space, the MC schedule is called Hamiltonian Monte Carlo. The Hamiltonian Monte Carlo execute a Monte Carlo dynamic in auxiliary space. With current sample $X^{(t)}$. The HMC sample a momentum vector from an auxiliary distribution $V^{(t)} \sim \exp(-v^T M^{-1} v/2)$. The joint sample $(X^{(t)}, V^{(t)})$ updated by running a Hamiltonian Dynamics in joint space via

$$\frac{dX_t}{dt} = \frac{\partial H}{\partial V}, \frac{dV_t}{dt} = -\frac{\partial H}{\partial X}.$$

Here $H(x, v) = -\log p(x) + \frac{1}{2}v^T M^{-1} v$ is the Hamiltonian of such mechanical system. HMC has many advantage that it mixes well for high-dimensional targets, and travels in joints space thus not easy to be trapped in local minima. Leap frog integrator is usually a practical choice for numerical updates (Neal, 2011). To make Markov Chain detail balanced, additional Metropolis correction is also needed for a Hamiltonian proposal. In short words, HMC iteratively accepts new position and momentum pair $(\tilde{x}, \tilde{v})$ with rate $\min 1, \frac{H(\tilde{x}, \tilde{v})}{H(x, v)}$ where $(\tilde{x}, \tilde{v}) = LeapFrog(x, v)$ as approximated Hamiltonian proposal.

## D  PROOF OF THEOREM 1

We give the proof of Theorem 1 here. To begin with, we give a lemma to bound KL divergence with Fisher divergence as shown in Yamano (2021)

**Lemma 2.** *For fixed $q$, there exists a dimension-free positive constant $c$ such that for every distribution $p$ which is both integral and log-integral with respect to $q$, and $p$ has same support as $q$, we have*

$$\mathcal{D}_{KL} \leq \frac{c}{2}\mathcal{D}_F(p, q).$$

*proof of lemma.* For every 1st order smooth function $f$, assume both $|f|^2$ and $\|\nabla f\|_2^2$ are integrable with respect to $q$, the log-Sobolev's inequality (Gross, 1975) shows that there exist a dimension-free positive constant $c$, such that

$$\int |f|^2 \log|f| q(x)dx \leq c \int \|\nabla f\|^2 q(x)dx + \|f\|_2^2 \log\|f\|_2^2.$$

Here $\|f\|_2^2 = \int |f|^2 q(x)dx$. Replace $f = \sqrt{p/q}$, we have

$$LHS = \frac{1}{2}\int (p/q)\log(p/q)q = \frac{1}{2}\mathbb{E}_p \log(p/q) = \mathcal{D}_{KL}(p, q).$$

So we have

$$\nabla\frac{\sqrt{p}}{\sqrt{q}} = \frac{1}{2}\left[\frac{\frac{\nabla p}{\sqrt{p}}\sqrt{q} - \frac{\nabla q}{\sqrt{q}}\sqrt{p}}{q}\right] = \frac{1}{2}\left[\sqrt{\frac{p}{q}}\frac{\nabla p}{p} - \sqrt{\frac{p}{q}}\frac{\nabla q}{q}\right] = \frac{1}{2}\sqrt{\frac{p}{q}}\left[\nabla\log p - \nabla\log q\right].$$

Thus the first term in RHS is

$$c\int \|\nabla f\|^2 q(x)dx = c\int \|\nabla\sqrt{\frac{p}{q}}\|^2 q(x)p = \frac{c}{2}\int \|\nabla\log p - \nabla\log q\|^2 p$$

$$= \mathbb{E}_p\|\nabla\log p - \nabla\log q\|^2 = \mathcal{D}_F(p, q).$$

Note that $\|f\|_2^2 = \int |f|^2 q(x)dx = \int (p/q)q = \int p = 1$. We combine both sides to conclude

$$\frac{c}{2}\mathcal{D}_{KL}(p,q) \leq \frac{1}{2}\mathcal{D}_F(p,q) + 0.$$

So we have

$$\mathcal{D}_{KL}(p,q) \leq \frac{c}{2}\mathcal{D}_F(p,q),$$

where $c$ be another positive constant. □

The above lemma shows that KL divergence is upper bounded with Fisher divergence, which we are using to train the sampler. With above lemma, we can calculate mixing time for Langevin correction in proof below

*Proof.* Assume target satisfies

$$\lim_{\|x\|_2 \to +\infty} (\frac{|\nabla \log q(x)|_2^2}{2} - \Delta \log q(x)) = +\infty.$$

then their exits a constant $\lambda > 0$, such that Poincare inequality holds for each $f \in C^1(\mathbb{R}^d) \cap L^2(q)$ with $\mathbb{E}_q f = 0$ Theorem 4.3 in Pavliotis (2014)

$$\lambda\|f\|_{L^2(q)}^2 \leq \|\nabla f\|_{L^2(q)}^2.$$

Let $p_0$ denotes the ASS distribution, which is trained to be bounded with $\mathcal{D}_F(p_0,q) \leq \delta$. By lemma, the KL between initial distribution $p_0$ and target $q$ is bounded by Fisher divergence with a dimension-free constant $c$

$$\mathcal{D}_{KL}(p_0,q) \leq \frac{c}{2}\mathcal{D}_F(p_0,q) \leq \delta \leq +\infty$$

With Poincare's inequality holds, the KL along Langevin diffusion $dX_t = \nabla \log q(X_t)/2 + dW_t$ decays exponentially fast as in Theorem 4.6 in Pavliotis (2014)

$$\mathcal{D}_{KL}(p_t,q) \leq \exp(-2\lambda t)\mathcal{D}_{KL}(p_0,q)$$
$$\leq \exp(-2\lambda t)\frac{c}{2}\mathcal{D}_F(p_0,q)$$
$$\leq \exp(-2\lambda t)\frac{c}{2}\delta.$$

Thus if we want $\mathcal{D}_{KL}(p_t,q)$ to be controlled under tolerance $\epsilon$, we only need diffused time $t$ to satisfies

$$t \geq \frac{1}{2\lambda}\left[\log(\frac{c}{2}) + \log(\frac{\delta}{\epsilon})\right] = \frac{1}{2\lambda}\left[C + \log(\frac{\delta}{\epsilon})\right].$$

where we place $C = \log(\frac{c}{2})$ to be another constant. The diffusion time must be positive, thus we take

$$T^* = \max\{0, \frac{1}{2\lambda}\left[C + \log(\frac{\delta}{\epsilon})\right]\},$$

and finish the proof. □

# E EXPERIMENTAL DETAILS AND MORE RESULTS

## E.1 SYNTHETIC TARGET

For toy 2-dimensional data experiments, we use a 3-layer MLP neural network with 200 hidden units in each layer as the sampler. The activation of the sampler is chosen as LeakyReLU non-linearity with a 0.2 coefficient. The score network is a 3-layer MLP with 200 hidden units in each layer. The activation of the score network is GELU non-linearity.

When reporting the numbers in Tab 1, we compute MMD metrics based on a total of 2000 samples. We run 20 independent experiments for each target and algorithm to calculate the mean and standard deviation.

Figure 6 visualizes the model capabilities of FIS, AFIS, and AFIS+MC samplers for matching three 2-dimensional target energy functions.

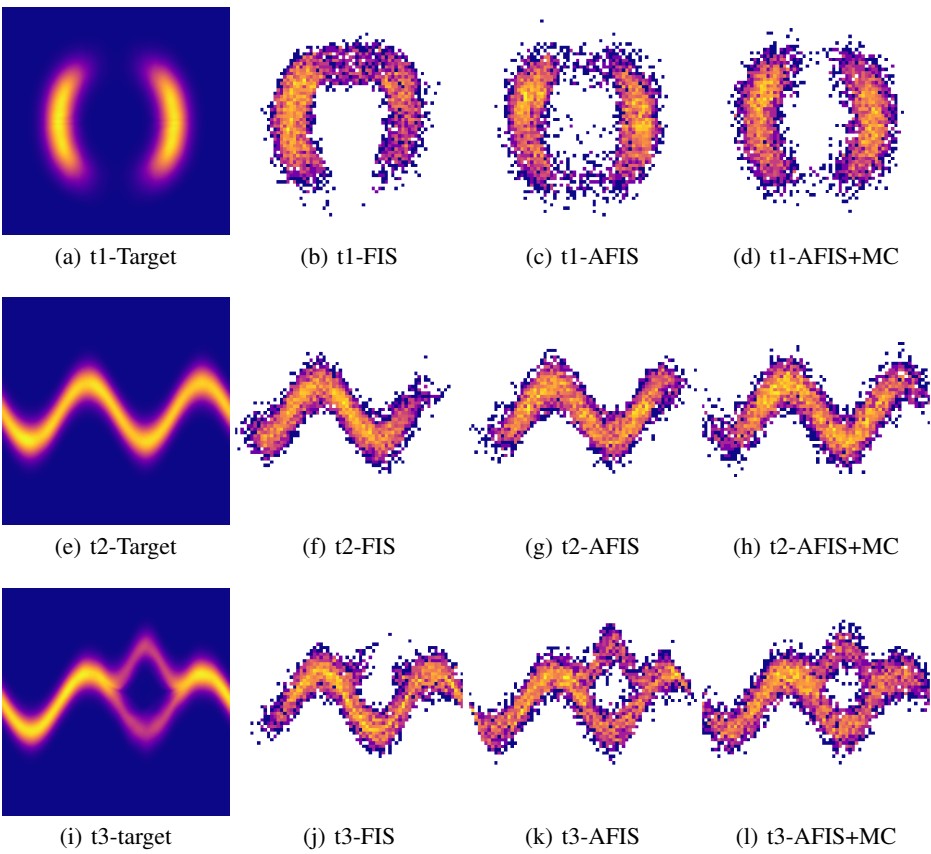

Figure 6: Comparison between samples generated by FIS, AFIS and AFIS+MC on three 2D energy functions.

## E.2 BAYESIAN REGRESSION

For high-dimensional data experiments, we also use 3-layer MLP neural networks as the sampler and score network, respectively. The activation of the sampler is chosen as LeakyReLU non-linearity with a 0.2 coefficient. The activation of the score network is GELU non-linearity. For Australian and Heart distributions, we use 400 hidden units in each layer and 600 hidden units for German distribution.

When reporting the numbers in Tab 2, we compute the MMD metric based on a total of 2000 samples. We run 20 independent experiments for each target and algorithm to calculate the mean and standard deviation. For the basic settings of Bayesian Regression problems, readers could refer to Song et al. (2017) for more details.

# F   FULL AFIS ALGORITHM

This section gives the full *Annealed Fisher Implicit Sampler* training algorithm.

---

**Algorithm 2:** Annealed Fisher Implicit Sampler training algorithm

---

**Input:** un-normalized target $\log q(x)$, annealed schedule $\{\lambda_k\}_{k=1}^K$, prior distribution
   $\log q_{prior}(x)$ ; latent distribution $p_Z(z)$, implicit sampler $G_\theta$, score network $s_\phi$,
   mini-batch size B, max iteration M.

Randomly initialize $(\theta^{(0)}, \phi^{(0)})$.

**for** *k in 1:K* **do**

  *# anneal the target*

  set $\log q_k(x) = \lambda_k \log q(x) + (1 - \lambda_k) \log q_{prior}(x)$

  **for** *t in 1:M* **do**

    *# update score network parameter*

    Get mini-batch from sampler $x_i = G_{\theta^{(t)}}(z_i), z_i \sim p_Z(z), i = 1, .., B.$

    Calculate score matching objective

$$\mathcal{L}_{SM}(\phi) = \frac{1}{B} \sum_{i=1}^{B} \left[ \|s_\phi(x_i)\|_2^2 + 2\langle \nabla_x, s_\phi(x_i) \rangle \right].$$

    Minimize $\mathcal{L}_{SM}(\phi)$ to get $\phi^{(t+1)}$.

    *# update sampler parameter*

    Get mini-batch latent code $z_i \sim p_Z(z), i = 1, \ldots, B.$

    Use re-parametrization trick to calculate S2D loss for sampler

$$\mathcal{L}_{S2D}(\theta) = \frac{1}{B} \sum_{i=1}^{B} \left[ \|\nabla_x \log q_k(G_\theta(z_i))\|_2^2 - \|s_{\phi^{(t+1)}}(G_\theta(z_i))\|_2^2 \right].$$

    Minimize $\mathcal{L}_{S2D}(\theta)$ to get $\theta^{(t+1)}$.

  **end**

**end**

**return** $(\theta, \phi)$.

---

