# OpenReview forum: "Annealed Fisher Implicit Sampler"
_ICLR.cc/2023/Conference — Submitted to ICLR 2023_

### Official Review · Reviewer_gNTU · 2022-10-21

**Confidence:** 4
**Correctness:** 3
**Technical Novelty And Significance:** 3
**Empirical Novelty And Significance:** 1
**Recommendation:** 3

**Clarity, Quality, Novelty And Reproducibility:**

The paper is clearly written. I had no issue following the math, the ideas, and the logic. The approach is relatively novel (a highlight of the paper) and if it performed well, would be a very interesting direction for further work and improvement. The paper as it is is not particularly reproducible. The paper lacks experimental details explaining the optimizers used, step sizes, batch sizes, training iterations, etc. Without these, I believe a reader would have a difficult time reproducing this work.

**Strength And Weaknesses:**

Strengths:

The method presented in the paper in interesting and the provided theory is detailed and appears correct. I was intrigued but the surrogate objective and its equivalence to the original goal of score matching. The theory was pretty easy to follow as well.


Weaknesses:

Overall the paper has 2 main weaknesses. First, I do not think the potential issues of the approach are explored in enough detail. Second the empirical evaluation is insufficient and lacking baselines.

Potential Issues:

I am concerned by utilizing an approximate to the score function of the model. This incorporates bias into the training objective. The authors show that when the score network is perfect then we are minimizing FD. How much error (in terms of FD maybe) is tolerable? Is it possible to bound the bias using the FD between the score network and implicit model? I am worried as well that this optimization may run into saddle point issues like with GANs or the neural stein sampler. It might be interesting to see how this approach works in a setting where we can get the exact score function (such as with a normalizing flow model). We could then compare how the normalizing flow performance works when trained with reverse-KL, true Fisher Divergence, and approximated Fisher Divergence (using the score network). Right now its not so clear how these things compare.

Evaluation:

Only two tasks are used for evaluation and these are relatively small scale in nature. Existing sampling approaches often present results on distributions with thousands of dimensions. I would be more convinced if the authors included at least one such distribution such as a pre-trained normalizing flow or a log gaussian cox process (two benchmarks which are becoming more standard in the field). Next the, paper is severly lacking baselines. In fact, there are no baselines beyond methods presented in this paper. The authors compare their method to the neural stein sampler. Why did they not compare with it? As well, the direct competitor to neural samplers are interactive methods such as MCMC and SVGD. We should also compare with these kinds of approaches. Sequential Monte-Carlo is currently the gold standard in the field and some such method should be used to give an indication of how well this approach performs to the best out there. Besides this there are a number of alternative neural sampling approaches such as the Path Integral Sampler which are beginning to gain traction. It would also be useful to provide one such comparison.


**Summary Of The Paper:**

This paper presents the annealed fisher implicit sampler which trains an implicit sampler (i.e GAN style model) to sample from an unnormalized probability distribution. The model is trained to minimize the Fisher divergence (i.e the score matching objective) between the model samples and the unnormalized distribution. This is an appealing objective for unnormalized distributions as the normalizing constant of the target is not necessary. However, we must be able to compute the the score function (i.e log-probability gradient) of the implicit model which is not tractable. The authors propose to use score matching techniques to learn a score network to approximate this quantity. They then propose a surrogate objective involving the target distribution, the score network, and the implicit sampler to minimize the fisher divergence.

The authors advocate for an annealed training objective which anneals the target from a simple distribution to the desired distribution which improves performance. Also the authors advocate to use a few steps of Langevin MCMC on top of the implicit sampler to correct for errors which they claim improves performance.

**Summary Of The Review:**

This paper proposes the Annealed Fisher Implicit Sampler which is a neural implicit sampling method training to minimize a novel objective  which approximates the fisher divergence between the target distribution and an implicit neural sampler. The method is interesting but the empirical evaluation is severely lacking. No baselines are presented besides methods proposed in this paper. As well, the method is not sufficiently explored and its problems (such as bias from score approximation) are not considered.

This method is interesting, but I believe a more thorough evaluation is necessary to understand how this approach will scale and how it compares to traditional sampling techniques and competing neural approaches.

---

> ### Author Response · Authors · 2022-11-16
> **Response to reviewer 4**
>
> Thank you for appreciation of our novelty. We summarize your major concerns and provide our responses in below.
>
> **Q1**. How much error (in terms of FD maybe) is tolerable? Is it possible to bound the bias using the FD between the score network and implicit model?
>
> **A1**. Firstly, we would emphasize the fact that the assumption of "perfect fitting" is very common in community with an assumption "with infinite capacity". Please refer to seminal works by Goodfellow [1,2]. Besides, the use of another 'perfectly matched' score network to aid an implicit generative model is also common in community [3]. Secondly, as in below, we give an error bound on our proposed S2D loss in terms of the L2 error of score estimation as you required. Formally, our error bound states.
>
> **Proposition 3**. Let $s_g(.)$ be the estimated score function of the implicit sampler. $s_g^*(.)$ be the true unknown score function. Assume these two assumptions hold:
>
> (1) The $L_2$-norm of the true score function of implicit sampler $s_g^*(x)$ is uniformly bounded $\|s_g^*(x)\|_2 \leq C_g$;
>
> (2) The estimated score function $s_g(x)$ is $\epsilon$-accurate under $L_2$, writes
>
> $$\mathbb{E}_{x\sim p_\theta} \|s_g(x)-s_g^*(x)\|^2_2 \leq \epsilon.$$
>
> here $s_g^*(.)$ is the unknown true score function of implicit sampler. Then the error of estimated loss function for generative process (i.e. S2D) will be at most $\max\{\epsilon(2C+1),\epsilon |2C-1| \}$, more specifically,
>
> $$|L(\theta) - \hat{L}(\theta)| \leq \max\{\epsilon(2C_g+1),\epsilon |2C_g-1| \}.$$
>
> Here
>
> $$\hat{L}(\theta) = \mathbb{E}_{x\sim p_\theta}\big[\|s_d(x)\|^2_2 - \|s_g(x)\|_2^2 \big]$$
>
>  denotes the estimated S2D loss with imperfect score estimation and
>
> $$L(\theta) = \mathbb{E}_{x\sim p_\theta}\big[\|s_d(x)\|^2_2 - \|s_g^*(x)\|_2^2 \big]$$
>
> is the perfect "unbiased" S2D loss as you mentioned.
>
> Both assumptions (1) and (2) are widely accepted across score-matching related literature ([4,5,6]).
>
> [1]Goodfellow I, Pouget-Abadie J, Mirza M, et al. Generative adversarial networks[J]. Communications of the ACM, 2020, 63(11): 139-144.
>
> [2]Arjovsky, M., Chintala, S., & Bottou, L. (2017). Wasserstein generative adversarial networks. In International conference on machine learning (pp. 214-223). PMLR.
>
> [3]Huszár, F. (2017). Variational inference using implicit distributions.
>
> [4]Lee, H., Lu, J., & Tan, Y. (2022). Convergence for score-based generative modeling with polynomial complexity.
>
> [5]Chen, S., Chewi, S., Li, J., Li, Y., Salim, A., & Zhang, A. R. (2022). Sampling is as easy as learning the score: theory for diffusion models with minimal data assumptions.
>
> [6]Nie, W., Guo, B., Huang, Y., Xiao, C., Vahdat, A., & Anandkumar, A. (2022). Diffusion Models for Adversarial Purification.

---

> > ### Author Response · Authors · 2022-11-16
> > **Response to reviewer 4 (2)**
> >
> > **Proof of proposition 3**.
> > $$\hat{L}(\theta) = \mathbb{E}_{x\sim p_\theta}\big[\|s_d(x)\|^2_2 - \|s_g(x)\|_2^2 \big]$$
> >
> > $$= \mathbb{E}_{x\sim p_\theta}\big[\|s_d(x)\|^2_2 - \|s_g(x)-s_g^*(x) + s_g^*(x)\|_2^2 \big]$$
> >
> > $$= \mathbb{E}_{x\sim p_\theta}\big[\|s_d(x)\|^2_2 - \|s_g(x)-s_g^*(x) + s_g^*(x)\|_2^2 \big]$$
> >
> > $$= \mathbb{E}_{x\sim p_\theta}\big[\|s_d(x)\|^2_2 - \|s_g^*(x)\|_2^2 - \|s_g(x)-s_g^*(x)\|^2_2 - 2\langle s_g^*(x), s_g(x)-s_g^*(x) \rangle  \big]$$
> >
> > $$= L(\theta) - \mathbb{E}_{x\sim p_\theta}\big[ \|s_g(x)-s_g^*(x)\|^2_2 + 2\langle s_g^*(x), s_g(x)-s_g^*(x)\big]$$
> >
> > Note that by Cauchy's inequality,
> > $$-\|s_g^*(x)\|_2 \|s_g(x)-s_g^*(x)\|_2 \leq \langle s_g^*(x), s_g(x)-s_g^*(x) \rangle \leq \|s_g^*(x)\|_2 \|s_g(x)-s_g^*(x)\|_2$$
> >
> > Recall assumptions (1) and (2), then we have
> > $$\|s_g^*(x)\|_2 \|s_g(x)-s_g^*(x)\|_2 \leq \sqrt{\epsilon} C_g$$
> >
> > So combine with the above equality, we have
> > $$L(\theta) - \hat{L}(\theta)= \mathbb{E}_{x\sim p_\theta}\big[ \|s_g(x)-s_g^*(x)\|^2_2 + 2\langle s_g^*(x), s_g(x)-s_g^*(x)\big] \leq \epsilon + 2\sqrt{\epsilon}C_g$$
> >
> > $$\epsilon - 2\sqrt{\epsilon}C_g \leq \mathbb{E}_{x\sim p_\theta}\big[ \|s_g(x)-s_g^*(x)\|^2_2 + 2\langle s_g^*(x), s_g(x)-s_g^*(x)\big]$$
> >
> > Thus we have
> > $$\|L(\theta) - \hat{L}(\theta)\|\leq \max\{\epsilon - 2\sqrt{\epsilon}C_g, \epsilon + 2\sqrt{\epsilon}C_g\}$$
> >
> > Note that in practice, the $L_2$ score estimation error $\epsilon$ can be arbitrarily small provided the unlimited capacity of score network. Thus the imperfect S2D loss can approximate the true S2D loss under any precision.
> >
> > **Q2**. I would be more convinced if the authors included at least one such distribution such as a pre-trained normalizing flow or a log gaussian cox process (two benchmarks which are becoming more standard in the field).
> >
> > **A2**. Both the pre-trained flow and log gaussian cox process are not admitted benchmarks. To show the power of our proposed sampler, we conduct an experiment to let our sampler learn to sample like a diffusion model trained on MNIST handwritten dataset. The diffusion model is a powerful generative model which learns the marginal score function of diffused data. The sampling of the diffusion model is slow, which requires thousands of iterations of SDE simulation. So we let our sampler learn with the score functions learned by the diffusion model. The results show that our proposed AFIS can learn from complicated diffusion models to sample on MNIST dataset. We put the comparison of samples from both diffusion model and our learned sampler in **section: distill diffusion model** in our supplement materials **afis_rebuttal.ipynb**.
> >
> > **Q3**. The authors compare their method to the neural stein sampler. Why did they not compare with it?
> >
> > **A3**. We do compare with the stein neural sampler in section 4.1. Quote :"The FSD-NS does not converge when training, so we omit the result of FSD-NS in comparison." Since the FSD-NS only partially update the sampler's parameters to minimize Fisher divergence, we empirically find that the FSD-NS is very hard to tune and diverges in most cases. So we omit the comparison with FSD-NS (need we compare a method which does not converge)? Besides, we report the implicit sampler trained with FSD-NS in (e) of Figure 4.
> >
> > **Q4**. The authors should also compare with these kinds of approaches.
> >
> > **A4**. The central benefit of implicit sampler is the efficiency. The implicit sampler only needs a single time Neural Function Evaluation to get a batch of samples, while traditional sampling methods such as MCMC require hundreds or thousands of iterations (or score function evaluations for score-based MCMC). So it is obvious that there is no necessity for the comparison of inference time between implicit sampler and iterative sampling methods. Since in our paper we are focusing on implicit samplers, we do not compare with explicit samplers such as discrete-time or continuous-time normalizing flows.
> >
> > To give a more comprehensive comparison of our proposed method, we conduct other experiments to compare with Hamiltonian Monte Carlo, a widely accepted standard MCMC method; the normalizing flow model, a representative method for explicit samplers. We put the comparison details in **section: comparison of afis, hmc and nf** in our supplement materials **afis_rebuttal.ipynb**..
> >
> > **Q5**. The paper lacks experimental details explaining the optimizers used, step sizes, batch sizes, training iterations, etc.
> >
> > **A5**. We discribe our neural architectures in Appendix E.1 and E.2. For all experiments, we use Adam optimizer with learnining rate lr=0.001, beta1=0.9 and beta2=0.999 as hyper-parameters. We will release our source code very soon.

---

### Official Review · Reviewer_mcoE · 2022-10-23

**Confidence:** 4
**Correctness:** 3
**Technical Novelty And Significance:** 3
**Empirical Novelty And Significance:** 3
**Recommendation:** 5

**Clarity, Quality, Novelty And Reproducibility:**

Generally, the paper is relatively clear. Notation-wise, the paper is very sloppy at times. E.g., on page 4, section 3.1, the authors introduce $s_d(x)$ without defining it, and the reader has to guess its meaning from the context. In section 2.2, $q_\phi$ is an approximation to $p$, whereas in other sections $p$ is sampler's approximation to the target $q$. The frequent change of notation $\log p(x)$ to $s_\theta(x)$ (being the same thing) and $s_\phi(x)$ (being approximation thereof) is also a bit confusing.

It is not very clear what MCMC correction step means in practice. How are they performed?

After the Proposition 2, it is mentioned that minimizing S2D loss is the same as Fisher Divergence. It is true only for a perfect score function estimation. In practice, it is approximate, so a mention of that is worthwhile in the bold text.

Typos:
- p. 13, in the definition of $l(x, f, \nabla f)$ -- an unnecesary integration sign.
- p. 14, second line of the second equation block: forgotten $dx$
- p. 16: messing up $p(\tilde{x} | x)$, $q(\tilde{x} | x)$

The authors do not provide any baselines or comparison against other methods besides their own. Even though the reported numbers (true posterior vs their method) are quite low, it would be beneficial to compare against competing methods, especially since the authors do cite them.

The authors provide experimental details, so reproducibility should not be an issue.

**Strength And Weaknesses:**

*Strengths*:
- The proposed method is theoretically well-backed. The loss enjoys the same gradient, as (intractable) Fisher divergence, provided a perfect estimate of the sampler's score function.
- The empirical study of the method's performance show that it indeed can capture target densities.

*Weaknesses*:
- The experimental results do not compare to any other methods.
- The clarity of the paper could use some improvements (concrete examples are given in the Clarity section).

**Summary Of The Paper:**

The authors propose a method to sample from a distribution with a known un-normalized posterior, named Annealed Fisher Implicit Sampler.
More specifically, they introduce the loss with the same gradient as the Fisher Divergence, provided a perfect estimate of the sampler's score function.
They train the sampler via alternatively estimating the sampler's score function and updating sampler parameters via the loss.
They employ annealing technique to alleviate sampler's training, with the rationale that the sampler is trained easily on easier target distributions.
They show the annealed sampler performance on a variety of benchmarks. They provide proofs of the propositions and a theorem, as well as experiment details in the appendix.

**Summary Of The Review:**

To sum up, the paper provides a novel method to sample from a distribution with a known unnormalized posterior. The contribution has potential impact, as sampling from such distribution has wide applications. The paper could use some improvement clarity-wise. It would be benefitial to provide comparison against other methods. Initial score: borderline.

---

> ### Author Response · Authors · 2022-11-16
> **Response to reviewer 3**
>
> Thank you for your patience in reviewing and collecting typos in our paper. We summarize the major concerns (**Q1**) in your reviews and our responses (**A1**) in below:
>
> **Q1**. The experimental results do not compare to any other methods.
>
> **A1**. In fact, in our paper, we compare AFIS to our main competing method, the Fisher Stein Discrepancy Neural Sampler (FSD), which is also an implicit sampler. As we show in Proposition 2, the FSD Neural Sampler only partially updates parameters aiming to minimize Fisher divergence. We found that FSD Neural Sampler fails in most cases in our experiments, resulting in diverging problems. So we omit the numerical performance of the FSD neural sample. However, as your kind suggestion, we also add another numerical experiment to compare three kinds of sampling methods.
>
> We compare the Hamiltonian Monte Carlo (HMC) [1-3] as a representative MCMC sampling algorithm baseline, the Normalizing Flow model trained with reversed-KL divergence as a baseline for explicit sampler [4,5], and our AFIS/FIS as a representative method for implicit samplers. To evaluate the quality of samples from each method, we calculate the Kernelized Stein Discrepancy (KSD)[6] between generated samples and target distribution. Since flow models are usually "large" models with massive parameters, we limit the flow model to approximately the same number of parameters as the implicit sampler. We find that our implicit sampler gives better samples than the flow model. The implicit sampler runs about 100 times faster than HMC method, which needs to run the Markov Chain iteratively. We put the experimental results below in **section: comparison of afis, hmc and nf** in our supplement materials **afis_rebuttal.ipynb**.
>
> Besides, we also add a distillation of an pre-trained diffusion model with the our sampler. Our diffusion model is trained on MNIST dataset. More details on distill diffusion model experiments can be found in responses to **section: distill diffusion model** in our supplement materials **afis_rebuttal.ipynb**.
>
> [1] Neal, R. M. (2011). MCMC using Hamiltonian dynamics. Handbook of markov chain monte carlo, 2(11), 2.
>
> [3] Neklyudov, K., Welling, M., Egorov, E., & Vetrov, D. (2020, November). Involutive mcmc: a unifying framework. In International Conference on Machine Learning (pp. 7273-7282). PMLR.
>
> [2] Neklyudov, K., & Welling, M. (2022, May). Orbital mcmc. In International Conference on Artificial Intelligence and Statistics (pp. 5790-5814). PMLR.
>
> [4] Dinh, L., Sohl-Dickstein, J., & Bengio, S. (2016). Density estimation using real nvp. arXiv preprint arXiv:1605.08803.
>
> [5] Rezende, D., & Mohamed, S. (2015, June). Variational inference with normalizing flows. In International conference on machine learning (pp. 1530-1538). PMLR.
>
> [6] Liu, Q., Lee, J., & Jordan, M. (2016, June). A kernelized Stein discrepancy for goodness-of-fit tests. In International conference on machine learning (pp. 276-284). PMLR.

---

### Official Review · Reviewer_G8ZU · 2022-10-23

**Confidence:** 4
**Correctness:** 3
**Technical Novelty And Significance:** 3
**Empirical Novelty And Significance:** 2
**Recommendation:** 5

**Clarity, Quality, Novelty And Reproducibility:**

Apart from Proposition 2, the paper is clearly written and easy to follow. However, the empirical evaluation is a bit weak to support the claim made by the paper. In terms of novelty, the proposed S2D seems to be novel to my knowledge.

**Strength And Weaknesses:**

## Strength
The paper is clearly written apart from the proof of Proposition 2. So in general, the paper is easy to read and understand. Although it is well-known that the Langevin sampler can be controlled by KL divergence and Fisher divergence is stronger (i.e. upper bounding) than other discrepancies, it is still good the formalize it in theorem 1. The proof strategy of theorem 1 is relatively easy to understand.

## Weakness
After reading the paper, I have several concerns: (1) the usefulness of the proposed approach; (2) the clarity of proposition 2, and (3) the empirical evaluation.

First, theoretically speaking, Fisher divergence is stronger than many of the well-known divergences and should provide better control over the convergence. However, in practice, it has been found that it is not the case. There have been many works investigating the theoretical properties of using Fisher divergence as the estimator (see [1], [2]). Especially in [2], it has been found that score matching may fail even under the simple student-t distribution. So, I wonder is your proposed approach works under the following unimodal setting:
1. Implicit sampler is a student t distribution with mean $\theta$
2. target distribution is another student t distribution with mean $\theta_0$, and $\theta$ and $\theta_0$ are far away from each other.
By minimizing the Fisher divergence, $\theta$ should approach $\theta_0$, is this true in practice (without using the MCMC step afterward)?


For proposition 2, I am not sure I understand it correctly. In the end of the proof, the equivalence is established with the $sg(\cdot)$ operator but in algorithm 1, the $\theta$ is back-propagated through the score estimator and target score. So why there isn't a $sg(\cdot)$ in this case?

For the empirical evaluation, the tasks are too simple to demonstrate the usefulness of the proposed approach. In the introduction, you mentioned the importance of the neural sampler, but I fail to see this based on the current empirical evaluation. For example, sampling from EBM is an interesting experiment since it is well-known MCMC sampling is too slow to train EBM. Another possible experiment is to train VAE where the encoder is the neural sampler. More baseline can also be considered, at least with the neural Stein sampler.




[1] Barp, A., Briol, F. X., Duncan, A., Girolami, M., & Mackey, L. (2019). Minimum stein discrepancy estimators. Advances in Neural Information Processing Systems, 32.

[2] Gong, W., & Li, Y. (2021). Interpreting diffusion score matching using normalizing flow. arXiv preprint arXiv:2107.10072.

**Summary Of The Paper:**

This paper proposed an implicit sampler by minimizing the Fisher divergence and then propose to improve it by using the annealing technique. Specifically, the author proposed an alternative objective such that minimizing it is equivalent to minimizing the Fisher divergence. Later, the author tried to solve the distant mode problem by creating a sequence of annealed distributions from simple Gaussian to target distribution. Theoretically, the author showed the equivalence of minimizing S2D (the proposed one) to Fisher divergence and also shows that the proposed implicit sampler can be used as an initialization for MCMC chains.

Empirically, the author conduct 1 synthetic experiment and 1 Bayesian regression to demonstrate the effectiveness of the proposed approach.

**Summary Of The Review:**

This paper provides an implicit sampler by minimizing Fisher divergence. My major concern is its usefulness due to the property of Fisher divergence and clarity of Proposition 2. Empirically, I think more large experiments should be conducted to see its effectiveness to support the claim made in the introduction.

---

> ### Author Response · Authors · 2022-11-16
> **Response to reviewer 2**
>
> Thank you for your careful and patient reading of our paper and proof details of proposition 2. We briefly summarize your main concerns (**Q1-3**) and provide our responses (**A1-3**) to them.
>
> **Q1**. The usefulness of the proposed approach and wondering about the success of AFIS (our proposed approach) without MCMC corrections under special target setting as studied in [1].
>
> **A1**. The concerns of failure of score estimation are studied in many literature. [1] studied the failure of score estimation under special target distribution, i.e., the student-t distribution. We believe the implicit sampler in our approach may not be bothered much with such concerns on score estimation.
>
> 1.Since we use a generic implicit sampler to approximate the target distribution in our proposed method, the score estimation is used to estimate the score function of implicit sampler instead of data directly drawn from special distributions such as student-t distribution. With suitable parametrization and approximation, the implicit sampler distribution can be not that much wired as the target.
>
> 2.Our implicit sampler is generic, so we do not need (also do not know how) to construct an implicit sampler with student-t distribution, thus the raised concern does not bother us.
>
> 3.Even if the target distribution is student-t itself, as you proposed in the second experiment setting you want us to check, our annealing technique (along with MCMC corrections) can avoid the direct score estimation for target distributions. We anneal the target distribution starting from a simple distribution, often a normal distribution. The implicit sampler then gradually learns to sample from annealed distributions. It is known that score estimation for simple distributions, i.e., normal distribution, is not problematic. Thus the sampler learn in a simple-to-hard manner, so as the score network. Such annealing method makes the sampler avoid being bothered by failure of direct score estimation on wired distributions. We also verify our words with another experiment in which we successfully learn a student-t distribution. We put the results in **section: student-t visualize** in our supplement materials **afis_rebuttal.ipynb**.
>
> **Q2**. For proposition 2, I am not sure I understand it correctly. In the end of the proof, the equivalence is established with the $\mathbf{sg()}$ operator but in algorithm 1, the $\theta$ is back-propagated through the score estimator and target score. So why there isn't a $\mathbf{sg()}$ in this case?
>
> **A2**. In Algorithm 1, $\theta$ is not back-propagated through the score estimation. Note that in the score estimation step of Algorithm 1, we only update the score network parameter $\phi$ with gradient-based methods, thus $\theta$ does not influence the score estimation for the score network (so naturally does not use the gradient of $\theta$). So we do not emphasize the stop-gradient on $\theta$ when doing score estimation. You may use a Pytorch style $x=G_\theta(z).detach()$ operation to stop the gradient of $\theta$ when doing score estimation. In fact, in our experiment implementation, we do stop the gradient of $\theta$ when doing score estimation as proposed in Proposition 2.
>
> **Q3**. The reviewer would like empirical evaluations for high dimensional distribution with our proposed method, for example, the distillation of a pre-trained EBM or using implicit sampler as VAE's implicit encoder.
>
> **A3**. Thanks for your nice suggestions. Firstly, we would like to emphasize that our evaluations on samplers, the 2D synthetic target, and bayesian inference are widely used for evaluating sampling algorithms as in [2-7]. Secondly, we additionally run an Diffusion model distillation experiment to demonstrate the use of our proposed method. The Diffusion Model (or score-based diffusion model) is a generative model which learns score function of data distribution along some diffusion precess. Since the diffusion model only provides score functions of "diffused" data distribution, more details in [8]. Our proposed AFIS/FIS naturally suitable for learning a sampler from knowledge of a pre-trained difffusion model. In our diffusion model experiment, we pre-train a diffusion model on MNIST handwrite dataset and teach a CNN sampler to generate like the pre-trained diffusion model. We learn the sampler by minimizing weighted Fisher Divergence as used in [8] and learn the sampler's score function with weighted denoising score matching technique. Our experiments shows that the sampler tends to learn the knowledge of pre-trained diffusion model. However, massive parameter tunning and engineering also need to be done, which we leave to our future study. We put the samples from pre-trained diffusion model and learned AFIS in **section: distill diffusion model** in our supplement materials **afis_rebuttal.ipynb**.

---

> > ### Author Response · Authors · 2022-11-16
> > **References**
> >
> > [1] Barp, A., Briol, F. X., Duncan, A., Girolami, M., \& Mackey, L. (2019). Minimum stein discrepancy estimators. Advances in Neural Information Processing Systems, 32.
> >
> > [2] Levy, D., Hoffman, M. D., & Sohl-Dickstein, J. (2017). Generalizing hamiltonian monte carlo with neural networks.
> >
> > [3] Song, J., Zhao, S., & Ermon, S. (2017). A-nice-mc: Adversarial training for mcmc. Advances in Neural Information Processing Systems, 30.
> >
> > [4] Wu, H., Köhler, J., & Noé, F. (2020). Stochastic normalizing flows. Advances in Neural Information Processing Systems, 33, 5933-5944.
> >
> > [5] Rezende, D., & Mohamed, S. (2015). Variational inference with normalizing flows. In International conference on machine learning (pp. 1530-1538). PMLR.
> >
> > [6] Neklyudov, K., Welling, M., Egorov, E., & Vetrov, D. (2020). Involutive mcmc: a unifying framework. In International Conference on Machine Learning (pp. 7273-7282). PMLR.
> >
> > [7] Neklyudov, K., & Welling, M. (2022). Orbital mcmc. In International Conference on Artificial Intelligence and Statistics (pp. 5790-5814). PMLR.
> >
> > [8] Song, Y., Sohl-Dickstein, J., Kingma, D. P., Kumar, A., Ermon, S., & Poole, B. (2020). Score-based generative modeling through stochastic differential equations. arXiv preprint arXiv:2011.13456.

---

### Official Review · Reviewer_enF9 · 2022-10-29

**Confidence:** 2
**Correctness:** 3
**Technical Novelty And Significance:** 3
**Empirical Novelty And Significance:** Not applicable
**Recommendation:** 3

**Clarity, Quality, Novelty And Reproducibility:**

The paper is not clear to me. Even the motivation about using Fisher discrepancy, and the various steps to correct it including annealing and then corrections.

I am unable to see the relevance to the community and the novelty.

i do not doubt the reproducibility

**Strength And Weaknesses:**


Weaknesses:
-- Some aspects of the paper are not clear to me. It seems these things are accepted in the community though. For example, why the need for a separate scoring network and the score estimation step in general ? If I am modeling p_\theta anyways, why can I not just use it directly? The authors jump between using s() and log p as the score function, which makes sense but the two notations and statements like "if we asynchronously estimate the sampler’s score function perfectly." tells me there is more to this that is not clear from the way it is written.

--

-- Please explain "asynchronous" estimation. When I hear "asynchronous", I tend to think of it in context of parallel or distributed optimization.

-- The derivative using s_d (x) below Figure 2 is not clear to me. What is s_d (x) ? Did I miss where it is defined ?

-- Minor: There are some typos and clarifications required, please proof-read .e.g.
"seems work" --> "seems to work".
"More specially"
What is \lambda in Prop 1 ?
Please mention what FSD stands for /before/ using it.


**Summary Of The Paper:**

The paper proposed a method to design a neural sampler that minimizes the fisher score discrepancy.

**Summary Of The Review:**

There are several aspects which are unclear to me, and hence I can not recommend acceptance. Further, I am worried about the relevance to the iclr community. I will reserve my judgement till rebuttals.

---

> ### Author Response · Authors · 2022-11-15
> **Response to reviewer 1 (1)**
>
> Thanks for your feedback. We briefly summarize your concerns (**Q1-9**) and post our responses (**A1-9**) for them. For the sack of logical continuity, we slightly adjust the order of raised concerns.
>
> **Q1**: If I am modeling $p_\theta$ anyways, why can I not just use it directly
>
> **A1**. The implicit sampler (or implicit model) uses a neural transform $G_\theta$ to push forward a random noise $z\sim p_{prior}$ to obtain a sample. The implicit sampler is favored for its high flexibility, for instance, neural transform can be Convolutional Neural Networks for image data or Transformer-based NN when dealing with sequence data. The sacrifice for such flexibility is that the implicit generative model does not has explicit probability formula. Typically, the used transform $G_\theta$ is neither bijective nor everywhere smooth, thus the explicit probability formula of a sample $x=G_\theta(z)$ is not tractable. For more knowledge of implicit generative models, please refer to existing works [1].
>
>
> **Q2**. Why the need for a separate scoring network and the score estimation step in general?
>
> **A2**. Since an implicit sampler does not has explicit probability formula. The training of it can not be avoided by relying on some other components. For example, the training of GAN (a famous implicit model and training paradigm) in seminal work [2] uses another neural network $D(x)$ to approximate the log-likelihood ratio of data and implicit model distributions. The training of WGAN in [3] uses another neural network to approximate the witness function for calculating the dual representation of Wasserstein distances. In our work, we aim to minimize the Fisher divergence between the implicit sampler-induced distribution and target energy-induced distribution. Thus it is unavoidable to use another network as also done in [1], [2], and other existing works. We choose to use a score network to approximate the score function of the implicit sampler and propose a tractable objective to minimize the Fisher divergence. We prove in theory that our proposed method is equivalent to minimizing the Fisher divergence under some conditions.
>
>
> **Q3**.The authors jump between using s() and $\nabla log p()$ as the score function.
>
> **A3**.Two notations for score function, the $s(.)$ and $\nabla \log p(.)$, are common in literature related to score functions. So we slightly abuse the use of two notions.
>
> **Q4**.Please explain "asynchronous" estimation.
>
> **A4**.Here we mention the words "asynchronous score estimation" to mean the alternative learning of score network to approximate the implicit sampler's score function. The score network and implicit sampler are learned alternatively with the other component fixed. We think another word "alternatively" may be more suitable from your view.
>
> **Q5**.The derivative using $S_d(x)$ below Figure 2 is not clear to me. What is $s_d (x)$ ?
>
> **A5**. Very sorry here for our typo. We abuse the use of $s_d(x) = \nabla \log q(x)$ without pre-defining.
>
> **Q6**.Please mention what FSD stands for /before/ using it.
>
> **A6**.We define FSD in section 2.1 for the first time it appears. Quote: "They called the above SD the Fisher Stein Discrepancy and the corresponding sampler FSD Neural Sampler."
>
> **Q7**. What is $\lambda$ in Prop 1 ?
>
> **A7**. $\lambda$ is defined in section 2.1, when introducing FSD Neural Sampler.
>
> **Q8**. The paper is not clear to me. Even the motivation about using Fisher discrepancy, and the various steps to correct it including annealing and then corrections.
>
> **A8**. We describe the motivation for using Fisher divergence in the first paragraph of section 2.1, quoting: "A general neural sampler does not have an explicit expression of the log-likelihood function, which we name them implicit samplers. Because of the un-normalized target distribution and unavailable log-likelihood, training implicit samplers by minimizing KL or related divergence always fails. An alternative way is to consider score-based divergence." We describe the need for annealing and corrections in the first paragraph of section 3.1. We provide both empirical evidence ((b,c,d) in Figure 4) and related works [4] in our paper, to show the necessity of the use of the annealing technique.
>
> **Q9**. I am unable to see the relevance to the community.
>
> **A9**. Our main contribution is to propose a tractable way to learn an implicit sampler that can sample from the target distribution. The implicit sampler learns the knowledge of target density and acts as an implicit representation of complicated target distribution. Such use of an implicit sampler or generative model to capture the implicit representation of a high-dimensional distribution is not rare in the ICLR community, please refer to existing accepted works in ICLR [5-8].

---

> > ### Author Response · Authors · 2022-11-15
> > **Response to reviewer 1 (2)**
> >
> > references
> >
> > [1] Mohamed S, Lakshminarayanan B. Learning in implicit generative models[J]. arXiv preprint arXiv:1610.03483, 2016.
> >
> > [2] Goodfellow I, Pouget-Abadie J, Mirza M, et al. Generative adversarial networks[J]. Communications of the ACM, 2020, 63(11): 139-144.
> >
> > [3] Arjovsky, M., Chintala, S., & Bottou, L. (2017). Wasserstein generative adversarial networks. In International conference on machine learning (pp. 214-223). PMLR.
> >
> > [5] Liu, B., Zhu, Y., Song, K., & Elgammal, A. (2020). Towards faster and stabilized gan training for high-fidelity few-shot image synthesis. In International Conference on Learning Representations.
> >
> > [6] Wang, W., Sun, Y., & Halgamuge, S. (2018). Improving MMD-GAN training with repulsive loss function. arXiv preprint arXiv:1812.09916.
> >
> > [7] Che, T., Li, Y., Jacob, A. P., Bengio, Y., & Li, W. (2016). Mode regularized generative adversarial networks. arXiv preprint arXiv:1612.02136.
> >
> > [8] Yu, S., Tack, J., Mo, S., Kim, H., Kim, J., Ha, J. W., & Shin, J. (2022). Generating videos with dynamics-aware implicit generative adversarial networks. arXiv preprint arXiv:2202.10571.

---

> > ### Comment · Reviewer_enF9 · 2022-11-16
> > **Thank you**
> >
> > I appreciate you taking the time to write the detailed response. But I do not think the paper is well-written enough to be ready for publication yet (I am not even commenting on other aspects of the paper). Please polish up the writing for clarity.

---

### Decision · Program_Chairs · 2023-01-20

**Decision:**

Reject

**Justification For Why Not Higher Score:**


The empirical study is not sufficient to demonstrate the advantages of the proposed method.

I encourage the authors to consider the suggestions provided by the reviewers, especially the suggestions on the experiments, to improve the quality of the paper.

**Justification For Why Not Lower Score:**

N/A

**Metareview: Summary, Strengths And Weaknesses:**


In this paper, the authors proposed to learn an implicit sampler by minimizing the Fisher divergence. To make the optimization tractable, the score function is then parametrized and learned. To further improve the performence, correction step is also utilized.

Most of the reviewers believe this method is interesting.

However, the major concern of the paper is the empirical study. The experiment part does not provide enough evidence to demonstrate the claimed benefits: 1, the competitors are lacking (there have been plenty of sampling methods proposed for unnormalized density, e.g., kernel Stein's method, Langevin dynamics, MCMC, et al,); 2, the applications are restricted to synthetic and Bayesian inference. It will be better if the method can be demonstrated on EBMs.